# Peculiarities of Pore Water Ionic Composition in the Bottom Sediments and Subsea Permafrost: A Case Study in the Buor-Khaya Bay

Alexander Ulyantsev [1,*], Natalya Polyakova [2] , Ivan Trukhin [2] , Yulia Parotkina [2] , Oleg Dudarev [3] and Igor Semiletov [3]

1   Shirshov Institute of Oceanology, Russian Academy of Sciences, 117997 Moscow, Russia
2   Institute of Chemistry, Far Eastern Branch of the Russian Academy of Sciences, 690022 Vladivostok, Russia; nvpolyakova@mail.ru (N.P.); truhin.ivan.91@gmail.com (I.T.); azarova.87@mail.ru (Y.P.)
3   Il'ichev Pacific Oceanological Institute, Far Eastern Branch of the Russian Academy of Sciences, 690041 Vladivostok, Russia; dudarev@poi.dvo.ru (O.D.); ipsemiletov@alaska.edu (I.S.)
*   Correspondence: uliantsev.as@ocean.ru

**Abstract:** This paper emphasises an ionic composition of the pore water of bottom sediments and subsea permafrost as an indicator of salinization of the thawed strata. Based on measurements of concentration of sodium ($Na^+$), potassium ($K^+$), calcium ($Ca^{2+}$), and magnesium ($Mg^{2+}$) cations, chlorides ($Cl^-$) and sulphates ($SO_4^{2-}$) in water extracts from bottom sediments and subsea permafrost deposits from three boreholes, a spatial difference in salinization of thawed strata within the Buor-Khaya Bay was shown. The vertical pattern of the macroions in the unfrozen segment was formed under subsea thawing of permafrost. The frozen strata contain fresh pore water and have been evolving under downward penetration of salt and subsequent thawing of subsea permafrost. Based on the analyses of thawed deposits, it was shown that the maximum pore water salinity was observed in the horizons enriched with sand and plant detritus. Over the boundary of subsea permafrost in the Ivashkina Lagoon, the pronounced total ion concentration (up to 50 g/L of $Cl^-$) of pore water was observed. This segment consists of moss debris, which is characterised by high porosity. The moss layer promotes the accumulation of dissolved pore water compounds and subsequent thawing of the frozen sediments.

**Keywords:** arctic shelf; bottom sediments; subsea permafrost; drilling cores; pore water; ionic composition; principal component analysis

## 1. Introduction

Observing, at the present time, environmental changes within the Arctic region, emphasises the need for investigations into properties of this sensitive natural system. These changes trigger such natural processes as massive degassing of the seabed caused by subsea permafrost thawing [1–6], coastal thermoerosion [7–9], and mobilization of ancient frozen organic carbon (OC) pool [10–17]. The rising importance of the study of potential consequences of climate change is undoubted where the East Siberian Arctic shelf represents a convenient area for research, where the extensive emissions of methane can be explained by subsea earthquakes and subsequent fracturing of the seabed, thawing of subsea permafrost, and degassing from gas hydrates due to thermobaric changes [1–4,6].

Bottom sediments coupled with pore water form an integrated natural system, which determines the migration of dissolved inorganic and organic components, the concentration of earth elements, and diagenesis in sediments. The composition of pore water substantially develops under environmental factors (physical, chemical, and biogeochemical) and depends on sedimentary peculiarities. To understand the potential mechanisms of subsea permafrost thawing, data on the chemical composition of pore waters is useful and helps

to assess the peculiarities of sedimentary systems. The results considered in the present paper emphasise the relations between cryogenic properties, grain size, lithology, and the compositional pattern of macroions in the interstitial water of the bottom sediments and subsea permafrost from the coastal zone of the Laptev Sea.

## 2. Materials and Methods

Three drilling cores were obtained during polar expeditions in 2014 (1–14 April, the 1D-14 (drilled 2–4 April) and 3D-14 (drilled 11–12 April)) and 2015 (28 March–14 April, the 1D-15 drilled 5–10 April), which were organised by the Pacific Oceanological Institute FEB RAS (Vladivostok), the National Research Tomsk Polytechnic University (Tomsk), and the Melnikov Permafrost Institute SB RAS (Yakutsk), in collaboration with Lomonosov Moscow State University (Moscow) and Shirshov Institute of Oceanology RAS (Moscow). Sediment cores were collected in the Buor-Khaya Bay using drilling apparatus PBU-2 coupled with skidding machine MTCh-4. Collection sites are located southeast of the Lena Delta near the Bykovsky Peninsula at the Buor-Khaya Bay (Figure 1). General characteristics of the drilled wells are presented in Table 1.

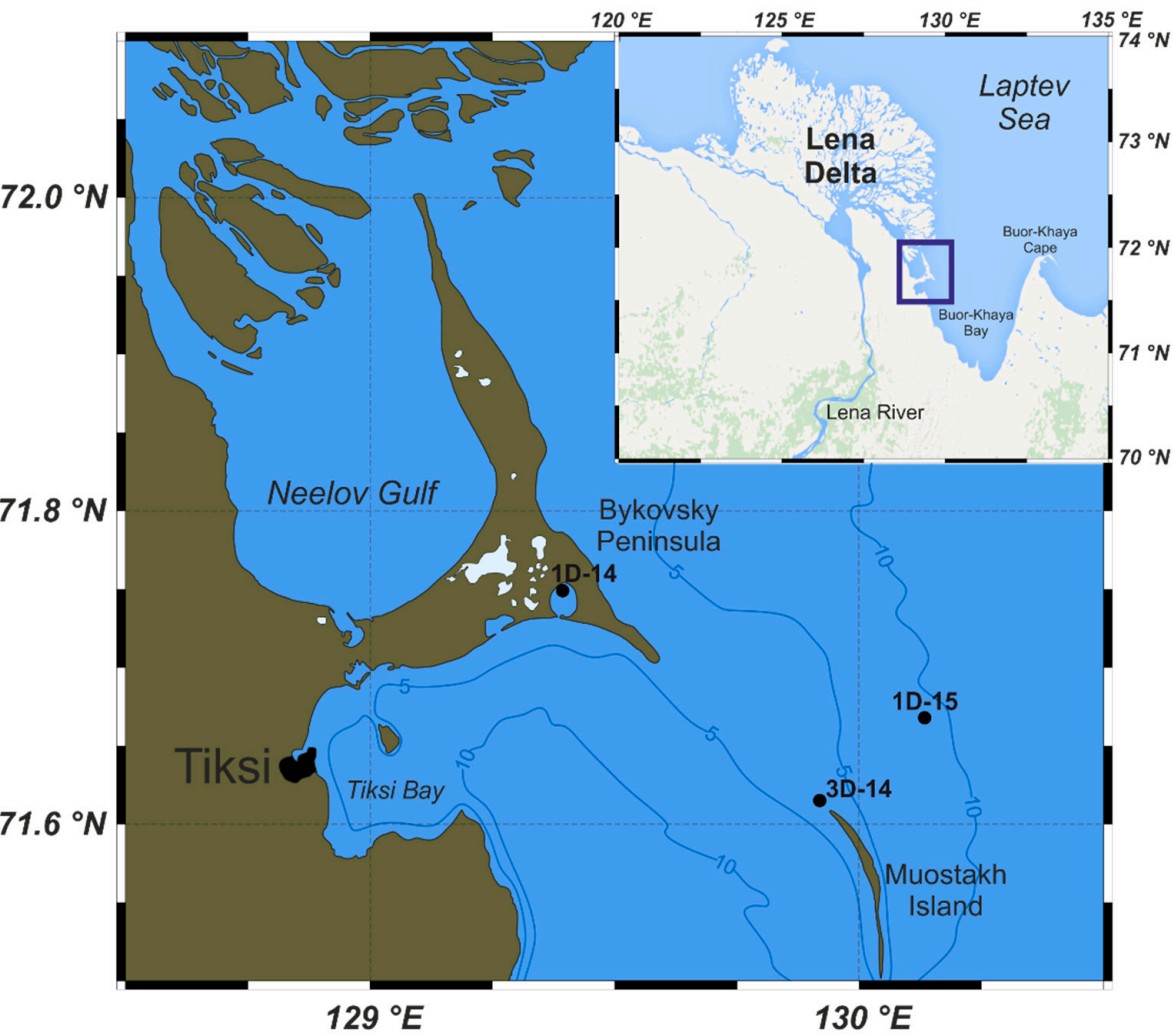

**Figure 1.** Location of the wells drilled in 2014 (1D-14 and 3D-14) and 2015 (1D-15).

**Table 1.** Characteristics of the drilled boreholes. Depth of the borehole is given from the sea bottom surface. Hereinafter the permafrost refers to the subsea ice-bonded permafrost.

| Borehole | Coordinates, °N; °E | Water Depth (Ice Thickness), m | Sediment Core Depth, m | Permafrost Table, m |
|----------|---------------------|-------------------------------|------------------------|---------------------|
| 1D-14 | 71.755; 129.397 | 3.1 (1.9) | 38.2 | 12 [1] |
| 3D-14 | 71.619; 129.916 | 2.7 (1.2) | 17.5 | 10 |
| 1D-15 | 71.672; 130.137 | 9.8 (1.6) | 33.2 | Not passed |

[1] The permafrost section of the 1D-14 core refers to the subsea ice-bonded permafrost.

The modern relief and coastal zone of the Bykovsky Peninsula was formed as a result of thermoabrasion, thermokarst, and sea-level changes [7,9,18–20]. The Ivashkina Lagoon of the Bykovsky Peninsula represents a thermokarst basin flooded with seawater, and its formation began between the Holocene and Pleistocene [21–23]. Located southeast of Bykovsky Peninsula, Muostakh Island, is a vanishing remain of the same plain and consists mainly of Late Pleistocene Ice Complex deposits [9,18]. Unfrozen sediments were sampled with a steel spatula after extracting the core. Sampling of consolidated permafrost deposits from the drilling cores was conducted with an electric screwdriver and rathole bits. All samples were put into polypropylene bags and kept frozen at −20 °C. A total of 105 samples were analysed. The numbers of samples were as follows: 38 samples of core 1D-14, 19 samples of core 3D-14, and 48 samples of core 1D-15.

The grain size of submarine deposits was measured in wet samples (approximately 20 g) using a Mastersizer 2000 particle analyser (Malvern Instruments, UK), according to ISO 13320:2009. The sand fraction was separated by wet sieving of samples on sieve of size 0.063 mm. The separated fractions were dried to a constant mass and weighed. The percentages of both silt and clay were determined after sieving on a particle analyser. The analytical conditions were 2000 rpm pump speed, 25 W ultrasound sonification power (40 KHz), 30 s exposition time for one measurement, and 2500 Hz scanning frequency. Milli-Q water was used as a dispersant and a blank. All laser diffraction analyses were done in triplicate and the results were averaged. The relative mass contributions (in %) of size fractions were expressed as follows: >63 μm, 10–63 μm, 2–10 μm, and <2 μm.

The ionic composition of pore water was studied by aqueous extraction. Wet sub-samples of deposits were dried with a MOC-120H moisture analyser (Shimadzu, Japan) at 105 °C for a constant mass. The dried samples (approximately 20 g) were placed into glass flasks and 150 mL Milli-Q water was added. Flasks with suspended samples were shaken with a Vortex Genius 3 shaker (IKA, Germany) for 15 min. Then, the water that was extracted was centrifuged using a 5702R centrifuge (Eppendorf, Germany) at 5500 rpm for 10 min to precipitate solids. The quantification of cations (sodium ($Na^+$), potassium ($K^+$), calcium ($Ca^{2+}$), magnesium ($Mg^{2+}$) and anions (chlorides ($Cl^-$) and sulphates ($SO_4^{2-}$)) was performed in supernatants. The moisture content in the sediments was expressed as gravimetric percent (%). The results of the ion concentration were recalculated by the volume of pore water in the subsamples. The pore water volume was calculated by the weight loss after the complete removal of water at 105 °C according to the bulk density of each subsample.

The identification of anions was carried out by ion exchange chromatography with conductometric detection using a Dionex ICS-5000 analyser (Thermo Scientific, Waltham, MA, USA) with pre-column and column PAX-100. A mix of carbonate buffer solution (3/4 mM of sodium carbonate/hydrocarbonate). Solution of acetonitrile (5%) was used as an eluent. The cations were quantified by atomic absorption spectrometry using a SOLAAR M spectrophotometer (Thermo Scientific, USA). The water extracts were diluted with Milli-Q water 100 times for sodium and potassium quantification, 20 times for calcium, and 10 times for magnesium resulting in an analytical range from 1 to 30 mg/L. Wavelengths were 766.5 nm for potassium, 589.0 nm for sodium, 422.7 nm for calcium, and 285.2 for magnesium, flame—mix of acetylene and air. Spectral and chemical noise

was eliminated by adding the buffer solutions: 0.1% of Cs for K and 0.2% of La for Ca and Mg. Precision for cations was at $\leq$10%. The results were recalculated considering a dilution rate and weight loss of the dried samples, and expressed as g/L.

## 3. Results

### *3.1. Concentration of Macroions*

The studied profiles were significantly heterogeneous in lithology and grain size and reflect the Holocene–Pleistocene history of sedimentation and sedimentary evolution in the examined area of the Laptev Sea shelf [24,25]. Various grain sizes of sediments with presence of the wood debris and herbal remnants, as well as pebble-gravelly associations showed the nearshore Laptev Sea submarine deposits to be of multiple origins and were formed under the influence of the Quaternary eustatic sea-level changes.

The concentration of $Na^+$, $K^+$, $Mg^{2+}$, and $Ca^{2+}$ in the pore water varies significantly along the strata, increasing at the permafrost boundary and reaching a minimum in the frozen horizons (Table 2). In general, changes in pore water salinity correlate with the cryogenic and geological structure of the sediment cores.

**Table 2.** Intervals of the ion concentrations and their mean values (under the intervals) measured in the pore water from thawed and frozen sections of the 1D-14, 3D-14, and 1D-15 sediment cores.

| Ion | 1D-14 | | 3D-14 | | 1D-15 |
|---|---|---|---|---|---|
| | **Thawed** | **Frozen** | **Thawed** | **Frozen** | **Thawed** |
| $Na^+$ | 6.25–25.2 / 13.8 | 0.01–0.89 / 0.28 | 2.92–10.0 / 6.04 | 0.16–3.80 / 0.81 | 3.33–12.7 / 6.32 |
| $K^+$ | 0.29–0.90 / 0.70 | 0.01–0.27 / 0.10 | 0.21–0.71 / 0.44 | 0.09–0.40 / 0.19 | 0.28–2.65 / 0.79 |
| $Ca^{2+}$ | 0.15–2.15 / 0.92 | 0.15–1.32 / 0.63 | 0.12–3.50 / 2.03 | 0.10–4.96 / 2.09 | 0.09–0.89 / 0.34 |
| $Mg^{2+}$ | 0.10–2.83 / 1.08 | 0.07–0.55 / 0.25 | 0.20–1.76 / 0.62 | 0.10–0.50 / 0.28 | 0.02–0.97 / 0.22 |
| $Cl^-$ | 6.79–49.7 / 20.5 | 0.02–1.36 / 0.45 | 6.05–13.4 / 9.82 | 0.04–3.50 / 0.85 | 3.30–19.9 / 9.38 |
| $SO_4^{2-}$ | 0.06–10.2 / 3.45 | 0.03–0.57 / 0.21 | 1.09–4.20 / 1.95 | 0.15–0.64 / 0.33 | 0.03–2.41 / 0.48 |

The 1D-14 profile varies drastically in ion concentration of pore water, especially in thawed strata (Figure 2) where deposits show negative temperature [3,24,26]. The temperature for the thawed section was not below $-1$ °C, while subsea permafrost differs in temperature below $-1.4$ °C. The concentrations of ions from the thawed section vary as follows: 6.25–25.2 (mean 13.8) g/L of $Na^+$; 0.29–0.90 (mean 0.70) g/L of $K^+$; 0.15–2.15 (mean 0.92) g/L of $Ca^{2+}$; 0.10–2.83 (mean 1.08) g/L of $Mg^{2+}$; 6.79–49.7 (mean 20.5) g/L of $Cl^-$; 0.06–10.2 (mean 3.45) g/L of $SO_4^{2-}$. The irregularity in ion concentration is considerable for sodium cations and chlorides. A significant salinity gradient is located above the permafrost boundary. The permafrost section differs in considerably lower concentrations of major ions in pore water which are comparable close to Yedoma Ice Complex deposits [23]. Values vary from 0.01–0.89 (mean 0.28) g/L of $Na^+$; 0.01–0.27 (mean 0.10) g/L of $K^+$; 0.15–1.32 (mean 0.63) g/L of $Ca^{2+}$; 0.07–0.55 (mean 0.25) g/L of $Mg^{2+}$; 0.02–1.36 (mean 0.45) g/L of $Cl^-$; 0.03–0.57 (mean 0.21) g/L of $SO_4^{2-}$.

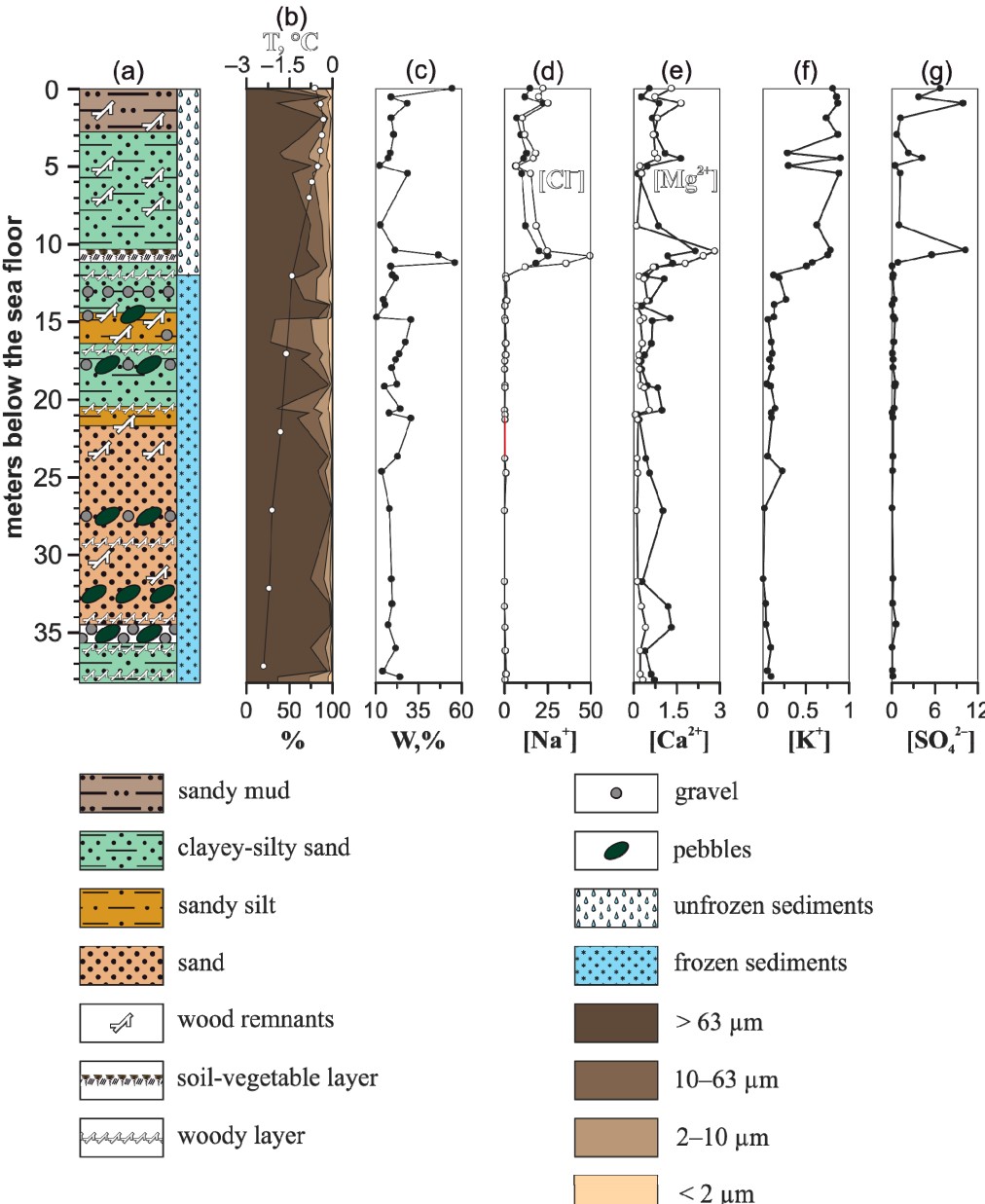

**Figure 2.** Geological characteristics and calculated concentration of ions in aqueous extracts of samples from the 1D-14 sediment core. (**a**) Lithology, (**b**) grain size with temperature (white circles), (**c**) gravimetric moisture content, (**d**) concentration of sodium cations (black circles) and chlorides (white circles), (**e**) concentration of calcium (black circles) and magnesium (white circles) cations, (**f**) concentration of potassium cations, (**g**) concentration of sulphates. The grain sizes are expressed as the percentage (%) of dry weight for each class. Concentration of ions is expressed in g/L.

The 3D-14 profile is comparably less variable in compositional patterns of ions. Ion concentrations here are considerably low as compared to 1D-14 (Figure 3). The thawed section varies in ions concentrations as follows: 2.92–10.0 (mean 6.04) g/L of $Na^+$; 0.21–0.71 (mean 0.44) g/L of $K^+$; 0.12–3.50 (mean 2.03) g/L of $Ca^{2+}$; 0.20–1.76 (mean 0.62) g/L of $Mg^{2+}$; 6.05–13.4 (mean 9.82) g/L of $Cl^-$; 1.09–4.20 (mean 1.95) g/L of $SO_4^{2-}$. The concentration of macroions of aqueous extracts from the permafrost section were as follows: 0.16–3.80 (mean 0.81) g/L of $Na^+$; 0.09–0.40 (mean 0.19) g/L of $K^+$; 0.10–4.96 (mean 2.09) g/L of $Ca^{2+}$; 0.10–0.50 (mean 0.28) g/L of $Mg^{2+}$; 0.04–3.50 (mean 0.85) g/L of $Cl^-$; 0.15–0.64 (mean 0.33) g/L of $SO_4^{2-}$.

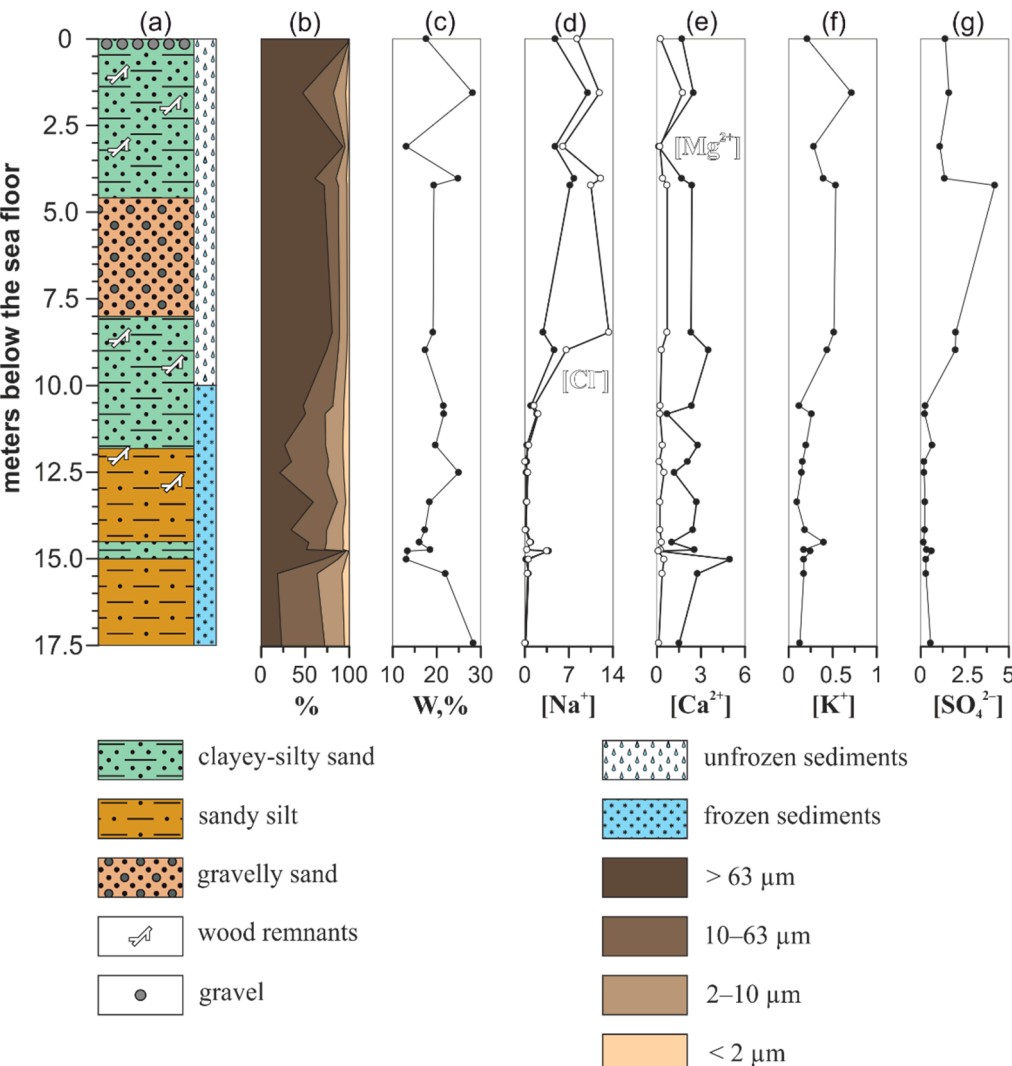

**Figure 3.** Geological characteristics and calculated concentration of ions in aqueous extracts of samples from the 3D-14 sediment core. (**a**) Lithology, (**b**) grain size, (**c**) gravimetric moisture content, (**d**) concentration of sodium cations (black circles) and chlorides (white circles), (**e**) concentration of calcium (black circles) and magnesium (white circles) cations, (**f**) concentration of potassium cations, (**g**) concentration of sulphates. The grain sizes are expressed as the % of dry weight for each class. Concentration of ions is expressed in g/L.

Unlike 1D-14 and 3D-14, the permafrost table at the 1D-15 sediment core site was not encountered and no pronounced concentration gradient of the ion concentration in the 1D-15 profile was found (Figure 4). Ion concentrations vary as follows: 3.33–12.7 (mean 6.32) g/L of $Na^+$; 0.28–2.65 (mean 0.79) g/L of $K^+$; 0.09–0.89 (mean 0.34) g/L of $Ca^{2+}$; 0.02–0.97 (mean 0.22) g/L of $Mg^{2+}$; 3.30–19.9 (mean 9.38) g/L of $Cl^-$; 0.03–2.41 (mean 0.48) g/L of $SO_4^{2-}$. The highest concentrations of $Na^+$, $Cl^-$, and $SO_4^{2-}$ were found in the upper section of 1D-15 profile (0–7 m), composed mainly of fine-grained sand, silt, and mud. The lower segment (27–33 m, horizons 2712, 2780, 2929, 3146, 3200, 3248, 3304 cm) is also characterised by an increased concentration of $Na^+$ and $Cl^-$, as well as $Ca^{2+}$ and $Mg^{2+}$, which could be caused by changes in the regimes of thalassogenic thawing of subsea permafrost sediments and/or sedimentation changes. The middle section of the studied 1D-15 profile, consisting mainly of sand with inclusions of clastic gravel-pebbles, is characterised by relatively low concentrations of pore water ions, as well as the lowest moisture content in the sediments. Considering the grain size of the middle pack of the 1D-15 profile, where plant detritus is absent, this sand section serves as a drainage

for downward seepage of seawater into the underlying strata, which ultimately leads to chemical thawing of the permafrost deposits.

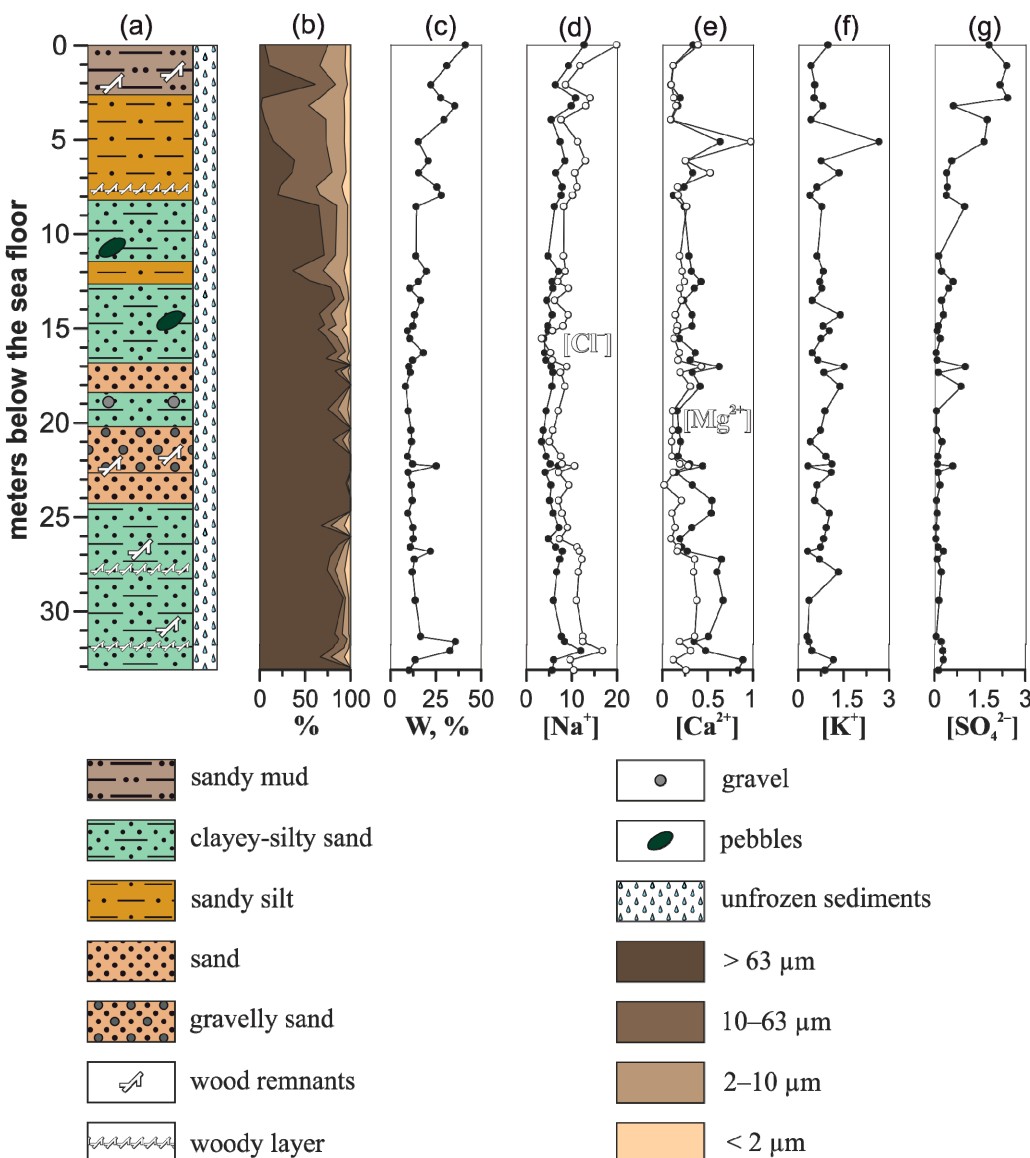

**Figure 4.** Geological characteristics and calculated concentration of ions in aqueous extracts of samples from the 1D-15 sediment core. (**a**) Lithology, (**b**) grain size, (**c**) gravimetric moisture content, (**d**) concentration of sodium cations (black circles) and chlorides (white circles), (**e**) concentration of calcium (black circles) and magnesium (white circles) cations, (**f**) concentration of potassium cations, (**g**) concentration of sulphates. The grain sizes are expressed as the % of dry weight for each class. Concentration of ions is expressed in g/L.

### 3.2. PCA Results

Principal component analysis (PCA) was used to identify the influence of grain size and moisture content in unfrozen sediment on the concentration of ions in the pore water. PCA results for the 1D-14 and the 3D-14 sediment cores have been reported earlier in [27] revealing differences in pore water composition between frozen and unfrozen sections. In the example of the 1D-15 sediment core, which was chosen due to absence of permafrost deposits, the PCA revealed the relationship between the ion concentration in pore water and the types of bottom sediments (Figure 5). The first two principal components (PC) explain more than 70% of the total variance in the data set. PC1 captures 45.1% of the

total data set variance, and PC2 explains 26.5%. The PC1 shows high loadings of $Na^+$, $Cl^-$, $SO_4^{2-}$, as well as moisture (W) and sand (>63 μm) contents, indicating that the above-mentioned parameters ($Na^+$, $Cl^-$, $SO_4^{2-}$, the moisture content, and sand concentration) provide significant contributions for the PC1, while PC2 exhibits high loadings of $K^+$, $Ca^{2+}$, and $Mg^{2+}$.

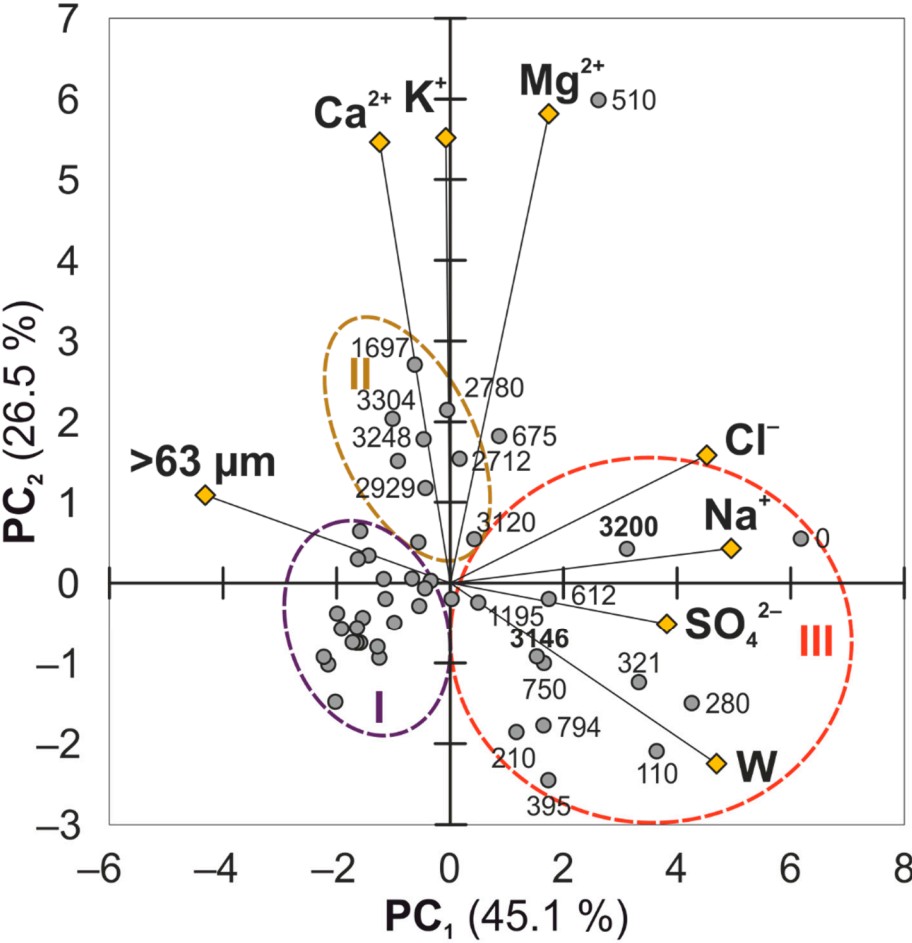

**Figure 5.** PCA biplot for the 1D-15 samples. W—gravimetric moisture content (in %); >63 μm—sand concentration (in %); $Na^+$, $K^+$, $Ca^{2+}$, $Mg^{2+}$, $Cl^-$, and $SO_4^{2-}$—calculated concentration of the following ions (in g/L) in the pore water of each sample.

The PCA biplot shows the antagonism of the sand concentration in deposits with the moisture content. The concentration of chlorides, sulphates, and sodium correlates to the sand content and confirms their accumulation in sediments with high moisture content. The sand samples form a dense cluster axis to the lower left in the PCA biplot (cluster I). These samples relate predominantly to lower and middle sections of the 1D-15 profile. They are characterised by a reduced salinity of the pore water and simultaneous low moisture content as well. Only two sand samples, 3146 and 3200, are displaced to the right and differ in their high contents of vegetable debris, resulting in the accumulation of the moisture and dissolved salts as well. The cluster II differs from other samples with elevated content in $K^+$, $Ca^{2+}$, and $Mg^{2+}$ and relatively moderate values of the other parameters. The sample 510 differs drastically due to its higher concentration of $K^+$, $Ca^{2+}$, and $Mg^{2+}$. The third cluster includes the samples mostly taken from the top section of the 1D-15 sediment core and these differ in high moisture content and salinity of the pore water.

## 4. Discussion

The compositional pattern of the ions in the pore water of the studied sediment cores from the Laptev Sea was formed under the thawing of the permafrost and subsequent downward penetration of the saline seawater and the diffusion of salts. The dynamics of these processes most likely depend on the climatic and hydrodynamic environment as well as the composition of the deposits. Enriched by saline pore water, sediments resulting from chemical thawing of submarine permafrost appear to be expanded within the entire submarine cryolithozone and develop differently depending on the thermohaline structure of the water masses (e.g., 1D-14 vs. 3D-14 profiles). A significant concentration of dissolved salts (up to 50 g/L of $Cl^-$) was found above the permafrost table in the Ivashkina Lagoon (1D-14 profile). This salinized section consists of moss that differs in high porosity and moisture content. Indeed, this can be explained by a combination of downwards salt diffusion and brine movement. Moreover, high drainage of sand furthers downward fluxes of saline pore waters within the strata as well. High salinities in Ivashkina Lagoon basin can be also caused by the concentration of salts beneath the lagoon ice cover in winter and spring. Depending on the lagoon inlet and bathymetry, water exchange between Ivashkina Lagoon and Tiksi Bay may be completely cut off in winter, enhancing the salt concentration effect. This phenomenon has been described for Polar Fox Lagoon located to the west of Ivashkina Lagoon [28].

The thawed deposits are characterised by fluctuations in the concentration of the macroions of the pore water, revealing heterogeneities in fluxes of marine water and distribution of ions in pore water. The irregular pattern of ionic composition implies seasonal changes in seawater fluxes to thawed strata and eventually depends on the physical and lithological peculiarities of the deposits. This seasonality can be affected, for instance, by sea ice conditions when strong salt exclusion occurs during sea ice formation [29]. Contrarily, permafrost strata differ in stable ionic profiles without obvious fluctuations revealing freshwater conditions during deposition. The observed pore water salinity gradients in the 1D-14 and 3D-14 sediment cores can be explained by downward spreading of marine water and subsequent salinization of thawed deposits. Indeed, in case of 1D-14 this saline maximum is related to a highly moisturised moss layer which is characterised by a high porosity and moisture capacity. Below the permafrost boundary concentration of ions begin to decrease drastically revealing freshwater or subaerial conditions of deposition. Due to permafrost thawing, these layers were salinized. In case of the 3D-14 core a saline gradient was also observed near the permafrost boundary, but it is weaker compared to 1D-14. The comparably low contents of analysed ions can be explained by the specific location of the 3D-14 borehole. This site is located near the north cap of Muostakh Island and is exposed to riverine currents of the Lena River. Subsequently, downward thawing of strata here was affected by spreading of light-salted waters. In contrast, Ivashkina Lagoon is commonly isolated from riverine currents and affected by marine flows.

According to the measured concentrations, the ions were sub-grouped into two clusters: the first consisted of $K^+$, $Ca^{2+}$, and $Mg^{2+}$, the second—$Na^+$, $Cl^-$, and $SO_4^{2-}$. The ions of the second group come to the interstitial water mainly with sea water and spread within the sediments according to their lithology. The concentrations of the cations from the first group do not depend on sand and moisture contents in the sediments. These ones can be originated from both freshwater and sea water, but their vertical distribution over the profiles differs markedly from those of the second group. Vertical distribution of $Na^+$ and $K^+$ differs significantly which may be due to the difference of enthalpies of KCl (18 kJ/mol) and NaCl (4.5 kJ/mol) dissolution, as well as of the salt diffusion coefficients for KCl and NaCl (0.20 and 0.15 $m^2$/s at 25 °C, respectively). This may determine the different transport of matter in the direction of the temperature gradient of the sedimentary column porous space for potassium and sodium chlorides. The correlation between $Cl^-$ and $Na^+$ suggests that NaCl provides that transport.

In this study, the accumulation of the sea salt at the permafrost boundary is only observed at the northern part of Ivashkina Lagoon (1D-14). This phenomenon is com-

monly not typical as from, for example, the heat and salt diffusion coupled modelling [30]. The north cap of Muostakh Island (3D-14) is likely more recently inundated and the 3D-14 profile is not located in a former thermokarst basin. Here, the Na$^+$ profile looks somewhat diffusive, whereas the Cl$^-$ profile looks diffusive near the permafrost boundary, but convective for the upper 8 m of the column. Thus, 3D-14 represents a more simplified subsea permafrost case. In case of 1D-14 core, the coarse-drained deposits can foster faster rates of salt transport to the permafrost boundary through density-driven water flow.

At the same time, the good drainage properties of the sands composing most of the studied sediment cores promote the vertical migration of salts to the permafrost table, causing subsequent thawing. Apparently, the major driving force of this phenomenon is a thermodynamic effect [31]. For instance, the Soret effect (thermic diffusion) occurs due to temperature gradient and promotes the transport of dissolved salts in sediments with their subsequent accumulation near the permafrost boundary. A gradient of the density and electroconductivity of the interstitial water appear to also play a role in redistribution of dissolved salts, resulting in a total gradient effect. On the one hand, the layer of plant detritus acts as a reservoir for the accumulation of salts originating from the lagoon water, but on the other, it distorts the balance of salinity in the sedimentary promoting the chemical thawing of permafrost.

## 5. Conclusions

The vertical pattern of the macroions in the unfrozen segment of three studied profiles from the Buor-Khaya Bay was formed under the influence of thalassogenic thawing of permafrost caused by changes in environmental conditions, water mass flows, and the grain size. The permafrost section of the studied profiles was formed under freshwater conditions and evolved under the thawing and consequent sea water spreading to the lower horizons causing the interstitial water to become enriched with dissolved salts. This phenomenon was probably prevalent in the areas of subsea permafrost development. Unlike the permafrost matrix, the unfrozen section of the studied profiles is more variable in concentrations of macroions of interstitial water, reflecting the heterogeneity of dissolved salt inputs and its distribution within the sediments.

We suggest that the phenomenon of sea salt concentration in mosses is primarily due to the peculiarities of plant detritus. Mosses are characterised by high moisture capacity, the value of which is significantly higher compared to mineral particles of deposits. This allows mosses to accumulate significant amounts of water, and the temperature gradient between thawed and frozen deposits facilitates the transport of dissolved salts to the moss layer and their concentration. As observed in the 1D-14 sediment core, the temperature gradient coupled with the presence of the moss layer has created such a salt reservoir. This moss layer could represent an older thermokarst basin ground surface which may have been subject to very shallow inundation in the early stages of lagoon formation as sea levels rose. In addition, the bottom-fast ice could inject salts in the highly porous moss, effectively trapping salt early in the lagoon's existence.

As revealed through the 1D-14 and 1D-15 profiles, the downward coarsening, polyfacial, and polygenetic origin of deposits supports other investigations of thermokarst formations, especially in terms of contemporary subsea permafrost degradation [32]. The data obtained in the present study is critical for modelling "the salt effect" on submarine permafrost thawing, as well as for reliably accounting for the "hydrate factor", which implies the contribution of hydrate methane to the global carbon cycle. The obtained materials confirm that salinization of continental sediments within the Laptev Sea shelf occurs from above due to flooding by the sea. Moreover, the distribution of salinity, as well as the concentration of elements [33] and organic biomarkers [34], correlate with the cryotic state of deposits. The lithology, grain size, and hydrological regime of the Buor-Khaya Bay are considered to be the most important factors controlling the accumulation of sea water salt into thawed sediments.

**Author Contributions:** Writing—original draft preparation, visualization, project administration, conceptualization, A.U.; formal analysis, validation, N.P.; investigation, I.T. and Y.P.; organization and field work, O.D. and I.S. All authors have read and agreed to the published version of the manuscript.

**Funding:** This research was supported through the Russian Scientific Foundation (grant no. 19-77-10044) within the framework of the state assignment of the Shirshov Institute of Oceanology RAS (no. 0128-2021-0005).

**Institutional Review Board Statement:** Not applicable.

**Informed Consent Statement:** Not applicable.

**Data Availability Statement:** The data are available online at the Mendeley Repository: http://dx.doi.org/10.17632/y6y6cg94dw.3 (accessed on 3 November 2021).

**Acknowledgments:** We thank Mikhail Grigoriev, Alexander Charkin, Vladimir Tumskoy, Andrey Koshurnikov, and the crew of Tiksi Hydrobase for their participation and help in the field research.

**Conflicts of Interest:** The authors declare no conflict of interest.

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
