# Peer review of "Peculiarities of Pore Water Ionic Composition in the Bottom Sediments and Subsea Permafrost: A Case Study in the Buor-Khaya Bay"

_geosciences, doi:10.3390/geosciences12020049_

Round 1

Reviewer 1 Report

Not surprisingly considering its history the publication presents important data for the three Artic cases studied. The authors show a strong interest in clearly communicating the data and its significance, which is appreciated. However, there is one fairly serious mistake. The same caption is attached to both Fig. 3 and Fig. 4. It should reference Figure 3 only with Fig. 4 reserved for 1D-15.

Starting from a low base, I feel *I have learned much from this manuscript. 

Author Response

Many thanks for reading the manuscript carefully! The mistake you mentioned has been corrected.

Reviewer 2 Report

Dear authors,

Thanks for the very interesting read! Please find attached my comments.

Author Response

Many thanks for reading our manuscript carefully and for your valuable recommendations, which made the article much better!

Our response can be found in the attachment.

Round 2

Reviewer 2 Report

Dear authors,

Overall, I am satisfied with the responses to my review. There are a still a couple of items that need clarification, but I expect this to be quite quick and I classify the current review status as "minor". I corrected the English thoroughly during the first review, but I would recommend a strong proof read after the next round of corrections are complete. I attach my responses to your responses, as well as a small list of minor points.

Thanks again for a very interesting read.

Author Response

Many thanks for your comments and suggestions. Our response is in attachment.
